# Determination of Selenium Speciation in High Se-Enriched Edible Fungus *Ganoderma lucidum* Via Sequential Extraction

**Wenyao Shi [1], Yuzhu Hou [2,†], Zezhou Zhang [3], Xuebin Yin [3], Xiaohu Zhao [4] and Linxi Yuan [1,*]**

[1]  Department of Health and Environmental Sciences, Xi'an Jiaotong-Liverpool University, Suzhou 215123, China
[2]  Jiangsu Bio-Engineering Research Center for Selenium/Advanced Lab for Functional Agriculture, Suzhou Institute for Advanced Study, University of Science and Technology of China, Suzhou 215123, China
[3]  College of Resource and Environment, Anhui Science and Technology University, Chuzhou 233100, China
[4]  College of Resources and Environment, Huazhong Agricultural University, Wuhan 430070, China
*   Correspondence: linxi.yuan@xjtlu.edu.cn
†   Present address: Technology Centre of Hefei Customs, Hefei 230022, China.

**Abstract:** Edible fungi are often used as an important selenium (Se) source to improve human daily Se intakes as food or Se supplements because of their strong ability to accumulate Se, in which *Ganoderma lucidum* was widely accepted in China. However, the existing Se speciation analysis methods such as protease K-based or trypsin-protease XIV-based, have low extraction rate and enzymatic hydrolysis rate (<30%) on the Se-enriched edible fungi, resulting that it is impossible to effectively evaluate Se transformation and safety of Se-enriched edible fungi. In order to improve the extraction rate and enzymatic hydrolysis rates, 12 extraction methods (combination of buffer solutions and enzymes) including 4 two-step extraction methods and 8 three-step extraction methods were applied to extract Se from high Se-enriched *Ganoderma lucidum* (Total Se content 245.7 μg/g in dry matter (DW)) in the present study. The results displayed that one three-step sequential extraction method as aqueous solution extraction-pepsin extraction-trypsin extraction performed the best, by which the total Se extraction rate could reach 65%, the total Se enzyme hydrolysis rate was 40%, and the Se speciation was revealed as Selenite (63.6%), SeCys$_2$ (20.1%), SeMeCys (14.8%) and SeMet (1.5%) in this high Se-enriched *Ganoderma lucidum*. This study offers a reliable and efficient method to evaluate the Se transformation and the Se safety in high Se-enriched edible fungi.

**Keywords:** *Ganoderma lucidum*; selenium speciation; pepsin extraction; trypsin extraction; LC-UV-AFS





## 1. Introduction

*Ganoderma lucidum* (*G. lucidum*) is one of the rarest Chinese medicinal materials and can be beneficial for the heart, liver, spleen, and lung [1]. Modern pharmacology proves that *G. lucidum* can regulate the immune system, fight tumors and increase longevity in humans [2–4]. Selenium (Se), which is one of the most important trace elements for humans, plays a significant role in the immune system, antioxidation and hormone regulation [5]. In order to combine both advantages of *G. lucidum* and Se, Se-enriched *G. lucidum* was produced [6], which had a better ability to sweep away superoxide and hydroxyl radicals than those without Se [7].

However, the previous studies mostly focused on the nutrition component variation during the Se enrichment process [8–10], and few have an eye on the speciation of Se for Se-enriched *G. lucidum*. Se mainly has two forms: organic including selenomethionine (SeMet), selenocystine (SeCys$_2$) and methylselenocysteine (SeMeCys) and inorganic including selenate and selenite [9], and humans absorb inorganic Se passively while absorbing organic Se actively due to different metabolic pathways [11]. Under biological transformation, inorganic Se forms would be transferred into organic forms, mainly as selenocysteine and selenomethionine. SeMet is essential for thyroid gland function and

plays important role in reproduction, DNA production, and protecting the body from infection. In addition, SeCys is the 21st proteinogenic amino acid, which are involved in lots of cellular and metabolic processes. Furthermore, organic Se can better increase the content of glutathione peroxidase and improve the activity of selenase and further stimulate the immune responses [12]. Hence, it is of great importance to figure out the chemical forms of Se in Se-enriched *G. lucidum*.

According to our pre-experiments (Unpublished data), the traditional procedure using chain-protease, protease K and protease XIV only extracted less than 12% Se fractions from the Se-enriched *G. lucidum*, which should be related to the thickened cell wall of *G. lucidum* under Se stress [10]. In another study [6], around 30% of organic Se from Se-enriched *G. lucdium* was extracted via national standard method of China [13]. Therefore, an effective method is needed to improve the Se extraction rate on Se-enriched *G. lucidum* to perform further studies.

Protease K, pepsin and trypsin were common enzymes to extract different forms of Se from Se-enriched yeast [14–16], while Protease XIV and Pronase E were often used for Se speciation extraction in nuts or bamboo shoots [7,17]. Thus, the objective of the present study is to find out an optimal extraction method via comparing 12 single-/multi-steps to reduce the loss of Se volatilization, prevent the variation of Se species, and ensure a decent extraction rate of selenoamino acids in Se-enriched *G. lucidum.*

## 2. Materials and Methods

### 2.1. Preparation of High Se-Enriched G. lucidum

The strain *G. lucidum* (Serial number: 5.0653) was purchased from China General Microbiological Culture Collection Centre (CGMCC), Beijing, China, and cultured in a Se-rich liquid medium (50 mg/L $Na_2SeO_3$, 1% corn powder, 2% glucose, 3% extraction of bran, 0.3% $KH_2PO_4$, 0.2% $MgSO_4 \bullet 7H_2O$, natural pH) with 5% inoculation amount under 35 °C and 200 RPM. After 3 days, the obtained fermentation broth was centrifuged for 10 min at 3000 RPM and the deposition as mycelia of *G. lucidum* was collected after washing in DI water 3 times to remove the attached materials. Next, the collected mycelia were frozen dried and stored in the refrigerator at −20 °C for later use. The total Se content in the present dried mycelia is $245.7 \pm 52.9$ μg/g.

### 2.2. Experimental Procedures

The Se-enriched *G. lucidum* samples in 2.1 were ground via ball milling and then be centrifuged (Supplementary File SI-1-1). The collected sediments were extracted with five different buffer solutions (Deionized water (DW), Tris-HCl buffer (pH = 2.1) (T-H/2.1), Phosphate buffer (pH = 7.5) (P/7.5), Tris-HCl buffer (pH = 7.5) (T-H/7.5) and Glycine-HCl buffer (pH = 2.1) (G-H/2.1) to determine the Se speciation in the supernatants and residues, respectively (Supplementary File SI-1-2). Meanwhile, the collected sediments in SI-1-1 were also extracted in water combined with 12 different sequential extraction methods to determine the Se speciation in different fractions (Supplementary File SI-1-3).

### 2.3. Determination of Total Se and Se Speciation

The total Se contents in the samples were analyzed by Hydride Generation Atomic Fluorescence Spectrometry (HG-AFS 9230) (Beijing Titan Instrument Co., Beijing, China) [18]. The national standard reference material GSV-1 (shrub leaves) was used in the present study with a recovery of 85.5–117.8% and a relative standard deviation (RSD) of 0.76%. The instrument detection limit was 0.08 μg/kg.

The extracted Se forms were analyzed via Liquid Chromatography (LC-20AB, SHIMADZU, Kyoto, Japan) coupled with Hydride Generation-Atomic Fluorescence Spectrophotometer (HG-AFS 9230) (Beijing Titan Instrument Co., Beijing, China), in which a Hamilton PRP X-100 anion exchange column (Hamilton, Reno, NV, USA) was used as the stationary phase. The detailed analysis processes were described in Yuan et al. (2013) [18]. The organic Se standards including selenocystine ($SeCys_2$), selenomethionine (SeMet), and

methylseleno-L-cysteine (MeSeCys) were purchased from Tokyo Chemical Industry, Co., Tokyo, Japan, and the inorganic Se standard-selenite was booked from National Reference Material Centre, Beijing, China. The instrument detection limits and determination precisions for the Se species were 2, 10, 5, 2 µg/L, and 5%, 10%, 6%, 5%, respectively.

*2.4. Statistical Analysis*

All statistical analyses were performed using Microsoft Excel 2016, SPSS 19, and Origin 2018. The significant differences were calculated by a one-way ANOVA performed by SPSS 19 (at the significance level of $p = 0.05$).

## 3. Results and Discussion

*3.1. One-Step Extraction without Enzyme Has Low Extraction Rate*

In the one-step experiments, five different extraction solutions were selected to extract the high Se-enriched *Ganoderma lucidum*. As shown in Figure 1, the extraction efficiency under neutral conditions (Deionized water (DW), Phosphorous (pH = 7.5) (P/7.5), Tris-HCl (pH = 7.5) (T-H/7.5)) with 6–8% were slightly higher than those under acidic conditions (Tris-HCl (pH = 2.1) (T-H/2.1), Glycine-HCl (pH = 2.1) (G-H/2.1)) with 4–5%, but no statistic differences ($p > 0.5$). In addition, different buffer extraction solutions will bring different levels of Se loss during extraction with 20%, 17%, 16%, 13% and 2% in P/7.5, T-H/7.5, T-H/2.1, G-H/2.1 and DW treatments, respectively. Overall, the extraction rate using buffer solution alone was low (less than 10%), which was too low to become a suitable extraction method for high Se-enriched *Ganoderma lucidum*.

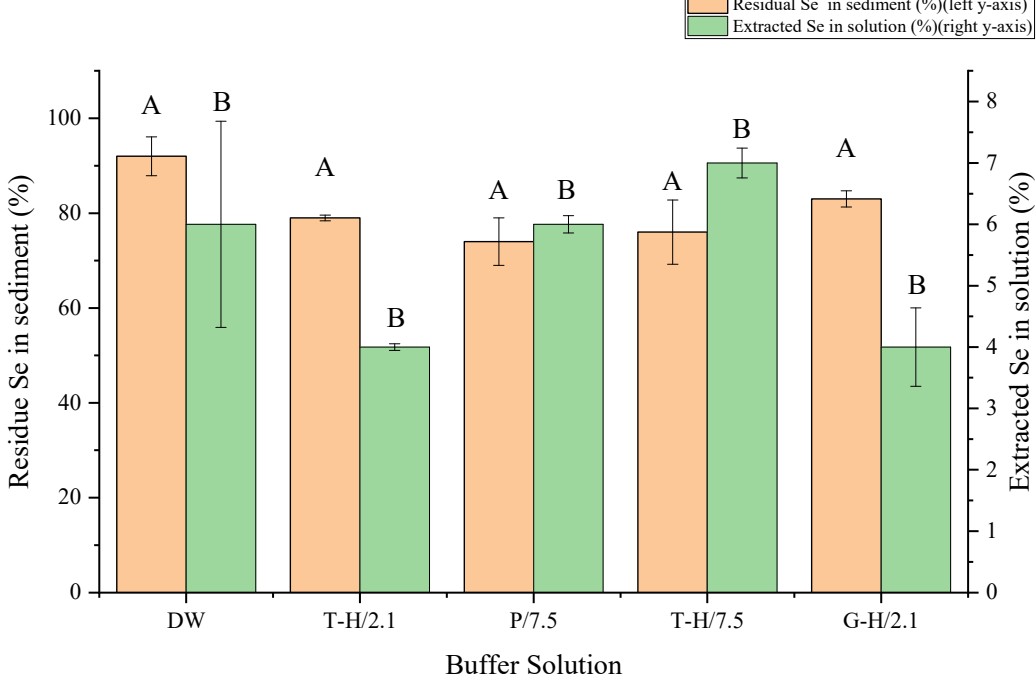

**Figure 1.** Extraction efficiency with 5 different buffer solutions. Note: DW-Deionized water; T-H/2.1-Tris-HCl (pH = 2.1); P/7.5-Phosphorous (pH = 7.5); T-H/7.5-Tris-HCl (pH = 7.5); G-H/2.1-Glycine-HCl (pH = 2.1). Different letters indicate significant differences at $p < 0.05$. Residue Se in sediment (%) = Total Se in sediment/Total Se in *Ganoderma lucidum* × 100%; Extracted Se in solution (%) = Total Se in supernatant/Total Se in *Ganoderma lucidum* × 100%.

Previous studies have been carried out to compare the extraction efficiency of one-step extraction on Se with four buffer solutions (Phosphate, pH = 7; Ascorbate, pH = 3; Water; Glycine/HCl, pH = 3) from broccoli [19]. Among four buffer solutions, the pH 7 phosphate buffer was summarized as the recommended one with around 30% Se extraction rate. While the extraction rate of Se with pH 7.5 phosphate in our study was 6%, which was obviously lower. This might be blamed on the cell wall structure differences between *Ganoderma lucidum* and broccoli, in which fungus is made up of a three-part matrix of chitin, glucans and proteins, while the main components of plants' cell walls are cellulose, hemicelluloses, pectin and agar [20]. Furthermore, due to chemisorption during Se fermentation process, Se can be bonded to the cell wall of fungi via ionic bonds or complexation, to thicken the cell wall [21], which were observed on high Se-enriched *Ganoderma lucidum* under Scanning Electron Microscopy (SEM) (Figure S1).

### 3.2. Sequential Extraction with Enzymes Increased Se Extraction Efficiency

As expected, the extraction rate of sequential extraction with enzymes (>50%) (Figure 2) is higher than that of one-step extraction without enzyme (<10%) (Figure 1) and the efficiency of complex-enzyme extraction (Methods E-L) (60–67%) is better than that of single-enzyme extraction (Methods A–D) (50–60%) (Figure 2). From the view of enzymatic hydrolysis rate, the extraction methods (E, F, G, H, I, J, K and L) with two enzymes involved had decent enzymatic hydrolysis rate at 30–40% while the enzymatic hydrolysis rate of single-enzyme extraction methods (A, B, C, and D) were at 20–30% (Figure 2). The Se loss rate of methods D, G, J, K and L were lower than 10%, which was comparatively lower than other groups (Figure 2).

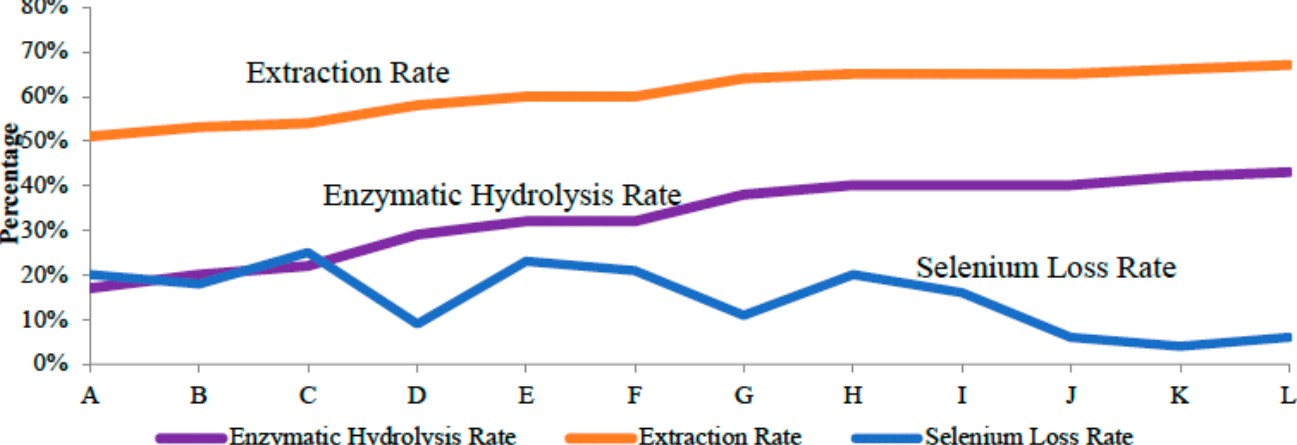

**Figure 2.** The extraction rate, enzymatic hydrolysis rate, and loss rate on Se for high Se-enriched *Ganoderma lucidum* under 12 sequential extraction methods. Note: A-Phosphate (pH 7.5), trypsin; B-Phosphate (pH 7.5), protease XIV; C-Tris-HCl (pH 7.5), trypsin; D-Tris-HCl (pH 7.5), protease XIV; E-Tris-HCl (pH 7.5), protease K; Phosphate (pH 7.5), protease XIV; F-Tris-HCl (pH 7.5), protease K; Tris-HCl (pH 7.5), protease XIV; G-Glycine (pH 2.1), pepsin; Phosphate (pH 7.5), trypsin; H-Phosphate (pH 7.5), protease K; Phosphate (pH 7.5), protease XIV; I-Phosphate (pH 7.5), protease K; Tris-HCl (pH 7.5), protease XIV; J-Glycine (pH 2.1), pepsin; Tris-HCl (pH 7.5), trypsin; K-Tris-HCl (pH 2.1), pepsin; Phosphate (pH 7.5), trypsin; L-Tris-HCl (pH 2.1), pepsin; Tris-HCl (pH 7.5), trypsin. Extraction rate (%) = Total Se in the extraction/Total Se in sample × 100%; Enzymatic hydrolysis rate (%) = Sum of total Se in extraction solution/Total Se of sediment (after ball milling) × 100%; Selenium loss rate (%) = (Total Se in *G. lucidum*—Se in extraction solution—Se in residue)/Total Se in *G. lucidum* × 100%.

### 3.3. Co-Application of Pepsin and Trypsin to Better Extract Se-Containing Amnio Acids

As showed in Figure 3a, selenite was detected for method B, D, E, F, H, I, K and L but only organic selenium for method A, C, G and J. The present inorganic selenium should be in the form of free selenite ion (2-) or untransformed sodium selenite. While

selenite (2-) is a conjugate base of hydrogenselenite and sodium selenite is easily dissolved in water [22,23]. Therefore, most of the selenite should be trapped into the water during the preliminary ball-milling process. Possible explanations for the existence of selenite could be a) artificial error: sodium selenite was not completely washed away during the preliminary step; b) Oxidation of organic selenium: SeMet could be transformed into SeCys and Se-Glu-Cys (ɣ-Glutamyl-Cysteine) while methane selenic acid and selenite would be the major oxidative products of SeCys and Se-Glu-Cys [24,25]. From this point of view, only method A, C, G and J should be taken into consideration.

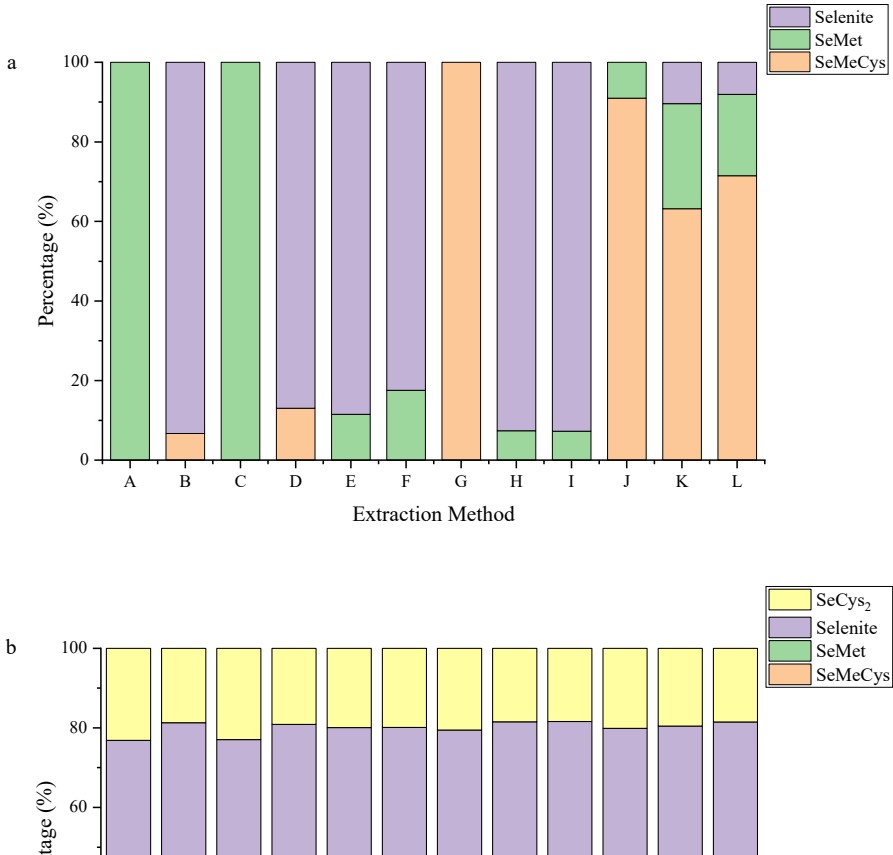

**Figure 3.** (**a**): Se speciation distributions in high Se-enriched *G. lucidum* via 12 sequential extraction methods (100% refers to total organic Se detected in sequential extraction); (**b**): Se speciation distributions (including the selenium speciation in the preliminary ball milling) in high Se-enriched *G. lucidum* (100% refers to total organic Se detected during the whole process) (Note: A-Phosphate (pH 7.5), trypsin; B-Phosphate (pH 7.5), protease XIV; C-Tris-HCl (pH 7.5), trypsin; D-Tris-HCl (pH 7.5), protease XIV; E-Tris-HCl (pH 7.5), protease K; Phosphate (pH 7.5), protease XIV; F-Tris-HCl (pH 7.5), protease K; Tris-HCl (pH 7.5), protease XIV; G-Glycine (pH 2.1), pepsin; Phosphate (pH 7.5), trypsin; H-Phosphate (pH 7.5), protease K; Phosphate (pH 7.5), protease XIV; I-Phosphate (pH 7.5), protease K; Tris-HCl (pH 7.5), protease XIV; J-Glycine (pH 2.1), pepsin; Tris-HCl (pH 7.5), trypsin; K-Tris-HCl (pH 2.1), pepsin; Phosphate (pH 7.5), trypsin; L-Tris-HCl (pH 2.1), pepsin; Tris-HCl (pH 7.5), trypsin.).

Organic selenium in yeast comes from the biotransformation of selenite with the help of catalyst [26]. Due to the very similar chemical physical properties between Se and S, the metabolism of them also has high similarities. Therefore, most Se-metabolites are analogous to S-metabolites [27]. For example, $SeCys_2$, SeMet and SeMeCys are analogues of cysteine, methionine and S-methylcysteine with Se in place of S [28]. However, $SeCys_2$ was transformed from serine instead of cysteine and SeMeCys can be produced from methionine [29,30]. Previous studies on amnio acid composition of *G. lucdium* indicated that the amount of serine (4.32–7.10%) is significantly higher than that of cystine (0.15–1.51%) and methionine (1.12–1.91%) in *G. lucidum* [31,32]. Hence, the amount of $SeCys_2$ in *G. lucidum* should be the highest. Considering that SeMeCys could be produced from methionine, the amount of SeMeCys should be higher than that of SeMet. Therefore, in terms of composition analysis, among method A, C, G and J, the composition of selenium in method J was most reasonable.

According to Figure 2, method G and J have lower selenium loss rate and higher extraction rate and enzymatic hydrolysis rate compared to method A and C. For method A and C, single enzyme extraction (trypsin) was used and only SeMet was detected but with low concentrations ($2.4 \pm 0.4$ µg/g, $3 \pm 0.7$ µg/g, respectively (Table S1)), indicating that trypsin could not extract selenite from the yeast when using alone. While for method G and J, sequential extraction method with pepsin and trypsin involved were used and the proportion of seleno amino acid significantly increased by around 10% compared with method A and C.

Protease K and protease XIV were used in method B, D, E, F, H and I and selenite became the major selenium form after sequential extraction, implying that protease K or protease XIV could make seleno amnio acid converting to form selenite. As for method K and L, selenite took a small proportion and Se-containing amnio acids tremendously increased compared to other methods, revealing that co-application of pepsin and trypsin could help the biosynthesis of organic selenium.

According to the component analysis in Table S1 and Figure 3a, both for methods B and D, protease XIV was used alone, and the speciation results were mainly inorganic Se and SeMeCys, indicating protease XIV might have the enzyme digestion site of SeMeCys. For method F, protease K was added before adding protease XIV and the resulting Se species were inorganic Se and SeMet in step 1. With the help of protease XIV from F-step 2, only one organic form as SeMet was detected. Meanwhile, protease K had been proved to cleave the peptide bonds containing the amino group of hydrophobic amino acid residues (such as L-leucine or L-tyrosine) [33]. Therefore, it can be indicated that protease K could cut SeMet sites. Furthermore, the structure or amino acid sequence of SeMet would be changed by protease K and therefore made protease XIV no longer can isolate SeMeCys from samples. Under this circumstance, Protease XIV was stimulated by the interaction between protease K and specific enzyme and therefore to be able to cut SeMet.

In Table S1, for method C with trypsin added, the only form of Se detected was SeMet. However, for method L-step 1 with pepsin involved, the speciation of Se was SeMet, SeMeCys, and selenite. While in L-step 2, after adding trypsin, there was only organic Se left and SeMeCys accounted for the main part. It can be inferred that trypsin not only can cut the SeMet site but also has the potential ability to cut the SeMeCys site. A combination of pepsin and trypsin would help to release more organic selenium forms rather than single enzyme [34]. Groups A and K showed similar results and could be explained by this inference.

Generally, proteases are divided into specific and non-specific, and different enzymes can hydrolysis different kinds of amino acids [35]. As showed in Table 1, existing finding on the functions of four enzymes were listed. All of them could cut peptide bonds but by attacking different terminals [36–38]. It was confirmed that the synergy between pepsin and trypsin hydrolysis existed, and therefore could enhance the efficiency of protein hydrolysis [39].

**Table 1.** Property of enzymes involved in this experiment.

| Enzyme | Medium | Origin | Function | References |
|---|---|---|---|---|
| Protease K | General | *Engydontium album* | cut peptide bond adjacent to the carboxyl group of aliphatic and aromatic amino acids with blocked alpha amino groups | [25] |
| Protease XIV | General | *Streptomyces griseus* | still under investigation | [26] |
| Pepsin | Acid | Gastric chief cell of stomach | cut peptide bonds between large hydrophobic amino acid residues and acts on proteins and converts them into peptones | [27] |
| Trypsin | Alkaline | PA clan superfamily in digestive system | cut peptide bonds at the C-terminal side of lysine or arginine and converts peptones into polypeptides | [27] |

The *G. lucidum* samples were treated with single enzyme and combined enzymes, respectively, and the differences in composition and quantity between the two methods were significant in the present study. *Streptomyces griseus* (Protease XIV), one of the non-specific proteolytic enzymes, has been proved to ensure a quantitative recovery rate when extracting Se from biological samples [40]. Subsequently, it was found that a better extraction rate was obtained when pepsin and trypsin were sequentially used compared with using pepsin, trypsin, or protease XIV alone [41,42]. Similarly, pepsin and trypsin were combined to treat the Se-enriched *G. lucidum* in the present study could cause a higher extraction rate and less Se loss in Figure 2.

*3.4. Se Species in Microorganisms*

Se speciation in different Se-rich materials were different. For example, SeMet is dominant in cereals and legumes [43–49], but SeMeCys is the major form in onions and broccoli [50,51]. Here we summarized the Se speciation data in micro-organisms published in literatures, especially in Se-enriched bacteria (probiotic) and fungus (mushroom and *Ganoderma lucidum*) (Table 2). The results showed the main Se species in *Agarcius bisporus* (160 µg/g), *Agarcius bisporus* (770.7 µg/g), *Pleurotus tuoliensis* and *Lentinula edodes* were SeMet, SeCys$_2$, SeMet and SeMeCys, respectively [41,52,53]. Barring the genetic difference, the enzymatic hydrolysis method and Se content of original yeast might be the key to explain the phenomenon. Similar to our research, single-enzyme-extraction (protease K) cannot fully release seleno amino acids in mushrooms, therefore only two organic forms of selenium were detected in *Pleurotus tuoliensis* [52]. Limited by the technical issue, only two organic Se species could be detected by Dernovic's group in 2002 [41]. Whereas contrast conclusion was achieved by Gergely [53] that SeCys$_2$ was the major organic selenium form in *Agarcius bisporus*. Possible explanation could be the high content of selenium in yeast made SeMet further transformed into SeCys$_2$ [54]. In addition, when the Se content in yeast is high, SeCys$_2$ usually becomes the major Se form instead of SeMet or SeMeCys, indicating that SeMet would have further transformation in a high-Se environment. Comparing Se extraction from Silage lactic acid and *Pediococcus acidliatici* [55,56], multi-enzyme-extraction could better release more species of Se. It was confirmed that inorganic selenium could be converted to nano-Se in yeast as well [57,58]. Whereas only one research listed above focused on the nano-Se detection [56]. The high content of nano-Se compared to other organic Se forms might be blamed on the inappropriate enzymatic hydrolysis method (single protease XIV) and therefore, other organic Se forms (SeMet, SeCys) could not be completely released. SeMet was the major Se form detected for the extraction from Lyophilized Se-enriched yeast and *Bifidobacterium bifidum* BGN4 [59,60], which could be put down to the incomplete enzyme extraction methods (single enzyme) as well. Protease

K and XIV are widely used to isolate seleno amnio acid as showed in in tradition. However, in this study, for *Ganoderma lucidum*, pepsin and trypsin could better cut the cleavage site on seleno amnio acid.

**Table 2.** Methods of Se isolation and speciation of Se in various micro-organisms.

| Sample Name | Total Se (µg/g) | Proteolytic Process | Se Speciation | References |
|---|---|---|---|---|
| *Pediococcus acidliatici* (bacteria) | 430 ± 4.0 | 1. Lysozyme 2. Protease XIV under untrasonic | SeCys$_2$ *, SeMecys, SeMet | [41] |
| *Agaricus bisporus* (mushroom) | 160 | 1. Water 2. Tris-HCl, lysing enzyme 3. Phosphate, protease XIV | SeMet *, SeCys$_2$, Selenite | [31] |
| *Pleurotus tuoliensis* (mushroom) | 100 | Protease K, water | SeMet *, SeCys$_2$, Se (IV) | [37] |
| *Lentinula edodes* (mushroom) | 46 ± 1.2 | 1. Tris-HCl 2. Trpsin 3. Protease XIV | SeMecys *, Se (IV), SeMet, SeCys$_2$ | [38] |
| *Agaricus bisporus* (mushroom) | 770.7 ± 37.3 | 1. Tris-HCl 2. Trpsin 3. Protease XIV | SeCys2 *, Se (IV), SeMet, SeMecys | [38] |
| Lyophilized Se-enriched yeast | 1300 | Tris-HCl, protease XIV | SeMet | [45] |
| *Bifidobacterium bifidum* BGN4 (probiotic) | 207.5 ± 1.25 | Phophate buffer, pronase E | SeMet | [46] |
| Silage lactic acid (bacteria) | 155.7 ± 9.7 | Protease XIV | nano-Se *, SeMet, SeCys$_2$ | [41] |
| *Ganoderma Lucidum* | 245.7 ± 52.9 | 1. Water 2. Glycine-Pepsin 3. Tris-HCl-Trypsin | Se (IV)*, SeCys$_2$, SeMet, SeMecys | this study |

Note: * refers to the majority selenium form, Se speciation rank from high to low (left to right), ± refers to standard deviation.

Selenite is the most common Se source to produce Se-enriched yeast [61]. First, selenohomocysteine (SeHCys) would be formed after selenite undergoing a series of physicochemical reactions [26]. SeHCys would be further transformed to become SeMet or selenocystathionine, of which SeMet could be converted to Se-adenosylselenomethionine (SeAM) or SeCys with the help of cystathionine c-lyase enzyme. Finally, reacting with S-adenosylmethionine (SAM) in the presence of selenomethyltransferase (SMT), SeCys could be converted to SeMeCys [62–65]. In the view of the mechanism of Se transformation in Se-enriched yeast, SeMet should be the major Se form. While for most of the yeast, the results seem counter intuitive. During the experiment, onion-like smell was noticed, and a possible explanation could be that SeMet was further methylated to form adenosyl homo-seleno cysteine (SeAHCys) and being released during the experiment [26].

## 4. Conclusions

In the present study, 12 extraction methods were designed to extract different Se species from Se-enriched *Ganoderma lucidum* with 245.7 ± 52.9 µg Se/g (DW) and the optimal sequential extraction processes with high extraction rate, low-transformed Se species, and low Se loss were screened as follows: Deionized water extraction/Pepsin extraction in 0.05 mol/L Glycine buffer (pH 2.1)/Trypsin extraction in 0.05 mol/L Tris-HCl buffer (pH 7.5). The screened sequential extraction method revealed that selenite (63.6%)

predominated in the high-Se-enriched *Ganoderma lucdium* ($245.7 \pm 52.9$ μg Se/g (DW)), and followed by $SeCys_2$ (20.1%), SeMeCys (14.8%) and SeMet (1.5%). In this study, LC-UV-AFS were used to separate four forms of selenium with good peak shape, reliable result and the method has good repeatability. Therefore, present screened sequential extraction method could be used to accurately determine the Se species in high Se-enriched edible fungus, vegetables and plant tissues, especially with Se accumulated in thickened cell walls and to evaluate the safety of Se-enriched edible fungus concerning the transformation of selenium.

There were unknown peaks detected during the experiment, indicating that some organic forms of Se were not recognized, which was limited by the types of Se species standards and the low amount of Se proteins in samples. Since $SeCys_2$ might be lost during the first water extraction step in the present study, a protocol should be proposed to protect $SeCys_2$ from being solved in water in the future study. Furthermore, a compressive Se speciation method is in urgent in that piles of Se forms could be identified through present method.

**Supplementary Materials:** The following supporting information can be downloaded at: https://www.mdpi.com/article/10.3390/horticulturae9020161/s1. Supplementary information including SI-1—Detailed experimental procedures, Figure S1—SEM (scanning electron microscope) photos on Se-enriched Ganoderma lucidum and normal Ganoderma lucidum and Table S1—Se speciation and their percentages in high Se-enriched Ganoderma lucidum via 12 sequential extraction methods, was included.

**Author Contributions:** W.S.: writing—original draft; visualization. Y.H.: formal analysis, investigation. Z.Z.: methodology. X.Y. and X.Z.: resources. L.Y.: conceptualization, supervision, project administration, writing—review and editing, data curation. All authors have read and agreed to the published version of the manuscript.

**Funding:** This research was funded by Key Laboratory of Se-enriched Products Development and Quality Control, Ministry of Agriculture and Rural Affairs/National-Local Joint Engineering Laboratory of Se-enriched Food Development (Se-2021B01), and Research Development Fund, Xi'an Jiaotong-Liverpool University (RDF-19-02-02).

**Institutional Review Board Statement:** Not applicable.

**Informed Consent Statement:** Not applicable.

**Data Availability Statement:** The data presented in this study are available upon request from the corresponding author.

**Conflicts of Interest:** The authors declare no conflict of interest.

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
