# Peer review of "Determination of Selenium Speciation in High Se-Enriched Edible Fungus Ganoderma lucidum Via Sequential Extraction"

_horticulturae, doi:10.3390/horticulturae9020161_

Round 1
Reviewer 1 Report
Comments and proposals for the authors:
It seems highly important to indicate in the Introduction the biological role of different Se forms for human
Comments:
1) Use Italics for the Latin names of mushrooms in the reference list
2) Use Journals abbreviations in the reference list
3) Provide Publishing house and place of publication for books in the reference list (Refs.1,2)
4) Use a larger font in Table to make the data perception easier
5) Add statistics in Figure 2
6) line 107 change ‘Fig.1’ to ‘Figure 1’
7) lines 206, 213, 218-220,252, 261,281 ‘SeMecys' change to ' SeMeCys,
8) line 272 Table 3 is absent in the text
Reviewer 2 Report
Research questions are well defined and within the aims and the scope of the journal. Material is accordingly defined. Methods are suitable, properly described and used in a way that is possible to replicate experiments and analyses. The investigation is performed to good technical standards. It is no ethical problem involved. Conclusions are well stated and based on the results, but too short. Discussion is sound and relevant.
Further suggestions:
Lines 67-68. What are other substances to reach 100% ?
In Figures it must be clear what refers to 100% each time.
Lines 246-247. Cited literature is not enough representative for »SeMet is dominant in cereals and legumes«, make here additional citations on SeMet in cereals, pseudocereals and legumes.
